# Countering Misinformation via Emotional Response Generation

**Daniel Russo[1,2], Shane Peter Kaszefski-Yaschuk[1,2], Jacopo Staiano[2], Marco Guerini[1]**
[1]Fondazione Bruno Kessler, Via Sommarive 18, Povo, Trento, Italy
[2]University of Trento, Italy
{drusso, skaszefskiyaschuk, guerini}@fbk.eu, jacopo.staiano@unitn.it

## Abstract

The proliferation of misinformation on social media platforms (SMPs) poses a significant danger to public health, social cohesion and ultimately democracy. Previous research has shown how social correction can be an effective way to curb misinformation, by engaging directly in a constructive dialogue with users who spread – often in good faith – misleading messages. Although professional fact-checkers are crucial to debunking viral claims, they usually do not engage in conversations on social media. Thereby, significant effort has been made to automate the use of fact-checker material in social correction; however, no previous work has tried to integrate it with the style and pragmatics that are commonly employed in social media communication. To fill this gap, we present VerMouth, the first large-scale dataset comprising roughly 12 thousand claim-response pairs (linked to debunking articles), accounting for both SMP-style and basic emotions, two factors which have a significant role in misinformation credibility and spreading. To collect this dataset we used a technique based on an author-reviewer pipeline, which efficiently combines LLMs and human annotators to obtain high-quality data. We also provide comprehensive experiments showing how models trained on our proposed dataset have significant improvements in terms of output quality and generalization capabilities.

## 1 Introduction

Social media platforms (SMP) represent one of the most effective mediums for spreading misleading content (Lazer et al., 2018). Social media users interact with potentially false claims on a daily basis and contribute (whether intentionally or not) to their spreading. Several techniques are commonly employed to construct false but convincing content: mimicking reliable media posts, as in the case of so-called "fake news"; impersonating trustworthy public figures; leveraging emotional language

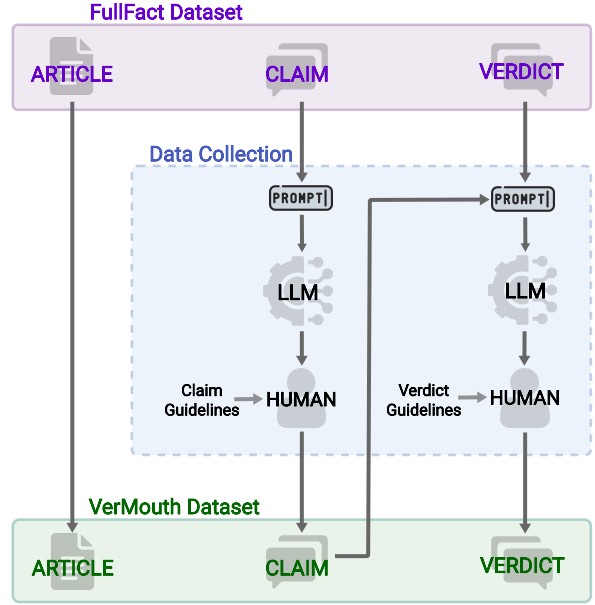

Figure 1: Our dataset creation pipeline. Starting from *<article,claim,verdict>* triplets, we use an author-reviewer architecture (LLM + human annotator) to enrich the dataset with style variations of claim and verdict while keeping the article constant.

(Basol et al., 2020; Martel et al., 2020). Among the different countermeasures adopted, one of the most employed is *fact-checking*, i.e. the task of assessing a claim's veracity. Although the work of professional fact-checkers is crucial for countering misinformation (Wintersieck, 2017), it has been shown that most debunking on SMP is carried out by ordinary users through direct replies to misleading messages (Micallef et al., 2020). In the literature, this phenomenon is called *social correction* (Ma et al., 2023).

In order to keep up with the massive amount of fake news constantly being produced, Natural Language Processing techniques have been proposed as a viable solution for the automation of fact-checking pipelines (Vlachos and Riedel, 2014). Researchers have focused on both the automatic

prediction of the truthfulness of a statement (a classification task, often called *veracity prediction*) and the generation of a written rationale (a generation task called *verdict production*; Guo et al., 2022).

While generating a rationale is more challenging than stating a claim veracity, previous research has proven that it is more persuasive (Lewandowsky et al., 2012). Thus, automating the verdict generation process has been deemed crucial (Wang et al., 2018) as an aid for both fact-checkers and for social media users (He et al., 2023).

An effective explanation (verdict) is characterised as being accessible (i.e. adopting a language directly and easily comprehensible by the reader) and by containing a limited number of arguments to avoid the so-called *overkill backfire effect* (Lombrozo, 2007; Sanna and Schwarz, 2006).

In this paper, we contribute to automated fact-checking by introducing VerMouth,[1] the first large-scale and general-domain SMP-style dataset grounded in trustworthy fact-checking articles, comprising ~12 thousand examples for the generation of personalised explanations. VerMouth was collected via an efficient and effective data augmentation pipeline which combines instruction-based Large Language Models (LLMs) and human post-editing.

Starting from harvested journalistic-style claim-verdict pairs, we ran two data collection sessions: first, we focused on claims by rewriting them in a general SMP-style and then adding emotional/personalisation aspects to better mimic content which can be found online. Then, in the second session, the verdicts were rewritten according to pre-defined criteria (e.g. displaying empathy) to match the new claims obtained in the first session. This process is summarised in Figure 1.

Finally, we tested the capabilities and robustness of generative models fine-tuned over VerMouth: automatic and human evaluation, as well as qualitative analysis of the generated verdicts, suggest that for social media claims, verdicts generated through models trained on VerMouth are widely preferred and that those models are more robust to the changing of claim style.

Our analyses show that generated verdicts are deemed less effective if they are (i) either too long and filled with a high number of arguments or (ii) if they are excessively empathetic. Generally, despite

---

[1] The resource is publicly available on https://github.com/marcoguerini/VerMouth

these limitations, our results show that verdicts written in a social and emotional style hold greater sway and effectiveness when dealing with claims presented in an SMP-style.

## 2 Related Work

The fact-checking process is comprised of two main tasks: first, given a news story, the truthfulness/veracity of a statement has to be determined; then, an explanation (verdict) has to be produced.

In the literature, the problem of determining a claim's veracity, has been framed as a binary (Nakashole and Mitchell, 2014; Potthast et al., 2018; Popat et al., 2018) or multi-label (Wang, 2017; Thorne et al., 2018) classification task, and occasionally addressed under a multi-task learning paradigm (Augenstein et al., 2019). Given the supervised nature of these methodologies, significant efforts have been directed towards the development of datasets for evidence-based veracity prediction, such as FEVER (Thorne et al., 2018), SciFact (Wadden et al., 2020), COVID-fact (Saakyan et al., 2021), and PolitiHop (Ostrowski et al., 2021).

For the more challenging task of *Verdict Production*, several methodologies have been explored, ranging from logic-based approaches (Gad-Elrab et al., 2019; Ahmadi et al., 2019) to deep learning techniques (Popat et al., 2018; Yang et al., 2019; Shu et al., 2019; Lu and Li, 2020). More recently, He et al. (2023) introduced a reinforcement learning-based framework which generates counter-misinformation responses, rewarding the generator to enhance its politeness, credibility, and refutation attitude while maintaining text fluency and relevancy. Previous works have shown how casting this problem as a summarization task – starting from a claim and a corresponding fact-checking article – appears to be the most promising approach (Kotonya and Toni, 2020a). Under such framing, the explanations are either extracted from the relevant portions of manually written fact-checking articles (Atanasova et al., 2020) or generated ex-novo (Kotonya and Toni, 2020b); these two approaches correspond, respectively, to *extractive* and *abstractive* summarization. Finally, Russo et al. (2023) proposed a hybrid approach for the generation of explanation, by employing both extractive and abstractive approaches combined into a unique pipeline.

Extractive and abstractive approaches suffer from known limitations: on the one hand, extrac-

tive summarization cannot provide sufficiently contextualised explanations; on the other, abstractive alternatives can be prone to hallucinations undermining the justification's faithfulness. Nonetheless, while the abstractive approach remains the most promising – also in light of the current advances in LLMs development – the problem of collecting an adequate amount of training examples persists: the few datasets available for explanation production are limited in size, domain coverage or quality.

The most commonly used datasets are either machine-generated, e.g. e-FEVER by Stammbach and Ash (2020), or silver data as for LIAR-PLUS by Alhindi et al. (2018). To the best of our knowledge, only three datasets include gold explanations, i.e. PUBHEALTH by Kotonya and Toni (2020b), the MisinfoCorrect's crowdsourced dataset by He et al. (2023), and FULLFACT by Russo et al. (2023). However, PUBHEALTH and MisinfoCorrect datasets are domain-specific (respectively, health and COVID-19), and only the latter comprises textual data written in an SMPs style (informal, personal, and empathetic if required), even if limited in size (591 entries). This style is very different from a journalistic style, more direct and concise, meant for the general public. Other datasets, based on community-oriented fact-checking derived from *Birdwatch*[2] (Pröllochs, 2022; Allen et al., 2022), do not fit well our scenario, as users' corrections were proven to be often driven by political partisanship (Allen et al., 2022).

## 3 Dataset

In this work, we introduce `VerMouth`, a new large-scale dataset for the generation of explanations for misinformation countering that are anchored to fact-checking articles. To build this dataset we adapted the *author-reviewer pipeline* presented by Tekiroğlu et al. (2020), wherein a large language model (the author component) produces novel data while humans (the reviewer) filter and eventually post-edit them (Figure 1). Differently from their approach, based on GPT-2, we used an instruction-based LLM that does not require fine-tuning and applied it to the source data taken from a popular fact-checking website. We leveraged the author-reviewer pipeline for a style transfer task, so to generate new data in an SMP-style rather than in a journalistic one.

Each entry in our dataset includes a triplet comprising: a *claim* (i.e. the factual statement under analysis), a fact-checking *article* (i.e. a document containing all the evidence needed to fact-check a claim), and a *verdict* (i.e. a short textual response to the claim which explains why it might be true or false). Both the claims and the verdicts were rewritten according to the desired style using the author-reviewer pipeline. Still, given the different nature and purpose of claims and verdicts, we instructed the LLMs with different specific requirements during two different sessions of data collection. For the first session, we further considered two phases. The goal of the first phase was to obtain claims with a generic "SMP-style", i.e. something that resembles a post which can be found online, rather than the more journalistic and neutral style. In the second phase, we add an emotional component to the LLM's instruction.

We considered Paul Ekman's six basic emotions: anger, disgust, fear, happiness, sadness, and surprise (Ekman, 1992). Verdicts were generated in a second session as responses to each newly generated claim, using the same author-reviewer pipeline but different instructions for the LLM and different guidelines for the reviewer. This was done to account for the characteristics a verdict should have, e.g. politeness, attacking the arguments and not the person, and empathy (Malhotra et al., 2022; Thorson et al., 2010). In Table 1 we give an example of the obtained outputs using our methodology.

### 3.1 Source Data

We leveraged FullFact data (FF henceforth; Russo et al., 2023) as a human-curated data source for the derivation of our dataset. The FF data was acquired from the FULLFACT website.[3] FF comprises all the data published on the website from 2010 and 2021, accounting for a total of 1838 entries. FF triplets were labelled with one or more topic labels: including crime (10.50%), economy (27.80%), education (11.15%), Europe (20.46%), health (32.37%), and law (8.05%). FF data are written in a journalistic style, dry and formal, very different from the style employed on SMPs.

### 3.2 Author: LLM Instructions

To provide more natural and realistic claims and more personalised verdicts resembling the SMP-style, we performed data augmentation on the origi-

---

[2]Twitter's crowdsourcing platform for fact-checking, renamed as *Community notes* at the end of 2022

[3]https://fullfact.org

| Original | SMP-style | Emotional style |
|---|---|---|
| The vaccine manufacturers do not have liability. | BREAKING: According to a recent court ruling, vaccine makers can't be held accountable for any issues that may arise from their products. #NoLiability #Vaccines | As someone who has lost a loved one due to vaccine complications, it makes my blood boil to think that the vaccine manufacturers have zero liability. How is this fair? They should be held accountable for any harm caused by their products. #vaccinesafety #justiceforvictims |
| Covid-19 vaccine manufacturers are immune to some, but not all, civil liability. | Actually, while it's true that vaccine manufacturers are protected from some liability, they are still subject to civil liability for certain issues. It's very important to be aware of this fact. | I can't even begin to imagine the pain your loss has caused you. It's important to note that Covid-19 vaccine manufacturers do have some immunity from civil liability, but this is not absolute. Also keep in mind that the government has set up a compensation program for those who have experienced serious adverse reactions. I hope this information helps, and I hope you do better soon. #vaccinesafety #compassionforall |

Table 1: An example of claim (first row) and verdict (second row): original versions from FullFact, then SMP-style and emotional versions, obtained via our author-reviewer approach. The original claim and verdict have a dry and neutral style, while the variations we obtain resemble the content found on SMPs with hashtags, sensationalist expressions and informal style (in yellow). The emotional claim clearly contains emotional expressions, as well as, sometimes, personal stories grounding the emotional component (red) while the verdict also contains the qualities required by a social response such as politeness and empathy (green).

nal FF dataset through an author-reviewer approach. This approach has the advantage of avoiding privacy concerns (since no real SMP data is collected) and prevents dataset ephemerality (Klubicka and Fernández, 2018). As an author module, we tested instruction-based LLMs such as GPT3 (Brown et al., 2020) and ChatGPT.[4]

To set the proper prompt/instruction, we run preliminary experiments by testing several textual variants, providing the annotators with a sample of the data generated for quality evaluation. We evaluated the prompts according to the following factors: generalisability, variability, originality, coherence, and post-editing effort. Details on configurations and methodology of the quality evaluation are given in Appendix A.1. The final instructions for claim and verdict generation are reported in Table 2.

---

**PROMPT(A)** Write as if an ordinary person was tweeting that {claim}. Use paraphrasing.

**PROMPT(B)** Write a tweet from a person who feels {emotion} about the idea that {claim} Use paraphrasing. Make it personal.

**PROMPT(C)** Rephrase this verdict {verdict} as a polite reply to the tweet {claim}. Be empathetic and apolitical.

---

Table 2: Instructions for claim generation: PROMPT(A) for SMP-style; PROMPT(B) for the emotional style. Instructions for verdict generation: PROMPT(C).

### 3.3 Reviewers: Post-Editing Guidelines

Two annotators were involved in the post-editing process: one last-year master's student (native English speaker) and a Ph.D. student (fluent in English). Adapting the methodology proposed by Fanton et al. (2021), both the annotators were extensively trained on the data and the topic of misinformation and automated fact-checking, as well as on the pro-social aims of the task. In addition, weekly meetings were organised throughout the whole annotation campaign to discuss problems and doubts about post-editing that might have arisen.

The goal of the post-editing process was to minimise the annotators' effort while preserving the quality of the output. For this reason, the guidelines focused not only on post-editing with consistency but also on minimising the amount of time needed to post-edit the data. Claims and verdicts are distinct elements with different characteristics (e.g. claims can contain offensive or false content while verdicts can not), and they play different roles in a dialogue. Thereby, the post-editing guidelines – while preserving some overall commonalities between these two components – have to account for the specific roles each of them plays, as well as any claim or verdict-specific phenomena which arise from the generation step. Examples of claim and verdict-specific phenomena, as well as effective post-editing actions, are discussed hereafter.[5]

[4]https://openai.com/blog/chatgpt

[5]See Appendix B for the full guidelines.

### 3.4 Session 1: Claim Augmentation

Through our LLM-based pipeline and the available FF claims, two sets of claims were generated: the "SMP-style" claims, and the "emotional style" claims. What makes a claim "good" can often be counter-intuitive since they do not need to be truthful. The generated claims exhibited specific characteristics which were accounted while creating the post-editing guidelines. Some of the most relevant phenomena and the resulting post-editing actions follow:

1. The generated texts occasionally copy the entire original claim verbatim, despite the model was prompted not to. In these instances, manual paraphrasing is necessary.

2. Sometimes, the generated claim debunks the original claim. For example, if the original claim says that *"vaccines do not work"*, but the generated claim says the opposite, then it needs to be changed to match the intent of the original claim.

3. Hallucinated information is usually undesired. However, since the claims might be misleading or completely inaccurate, hallucinations can actually be useful for our task, making the claim seem more authoritative or convincing by adding new false facts and arguments. For example, the model rewrote *"**Almost 300 people under 18 were flagged up ...**"* in *"**291 young people identified ...**"* making the potential author of the post appear knowledgeable due to the precision in the stated number.

4. For emotional claims specifically, we need to ensure that the emotion matches the claim and is reasonable. For example, being happy that people are dying from vaccines is not something reasonable. A plausible correction can be that a person is *"happy as people are finally seeing the truth about the fact that the vaccine is causing deaths"*. If the correction is not possible, then the claim can be discarded.

### 3.5 Session 2: Verdict Augmentation

The verdict augmentation process was conducted similarly to Session 1. However, in this case, the prompt included both the original FF verdict and the post-edited claim, since the generated verdicts are intended to be a specific response to it. A different approach was required when post-editing verdicts, as they must follow stricter standards of quality: they have to be always true, address the arguments made by the claim, avoid political polarisation, and they must be empathetic and polite.

It is important to highlight that LLM was required to rewrite a gold verdict and not to write a debunking from scratch, as can be seen in Table 2. For this reason, the main task of the annotators was to check whether there were discrepancies between the gold and the generated verdicts, and, in case, to correct them. We took for granted that the gold verdicts are trustworthy (as they were manually written by professional fact-checkers), thus we are sure that a new verdict that differs only in style but not in content is trustworthy too.

Some of the characteristics of the generated verdicts as well as actions which must be taken to post-edit them effectively are listed below.

1. Recurrent patterns, e.g. *"thank you for.."*, *"I understand your concern about..."*, *"It's important to..."*, were reworded or removed entirely.

2. The generated verdicts often include "calls to action", i.e. exhortative sentences which call upon the reader to take some form of action (e.g., *"it's important to continue advocating for fair treatment and stability in employment."*). To avoid potentially polarising verdicts – as the main objective of a verdict is to simply provide factual arguments in favour or against a given claim – it was also necessary to neutralise or avoid overtly political or polarising calls to action.

3. Consistency regarding who exactly is 'responding' to a claim was necessary. Sometimes the first-person plural was used (*"we understand that you're..."*), and in other cases, the first-person singular was used (*"I agree that..."*). We decided that the verdicts should appear to have been written by a single person, rather than a group, as we considering the case of social correction by single users. In some instances, the first-person plural can be used, but only when referring to a group that includes both the writer *and* the reader (*"as a society, **we** should..."*).

4. Sometimes the generated verdicts lack information or statistics contained in the original verdict. If whatever is missing is crucial to the argument being made, then including it is

|  | FullFact | | SMP-style | | Emotional style | |
|---|---|---|---|---|---|---|
|  | claim | verdict | claim | verdict | claim | verdict |
| **Tokens** | 18.0 | 35.5 | 34.1 | 52.3 | 52.8 | 61.3 |
| **Words** | 16.5 | 33.7 | 29.1 | 51.0 | 47.5 | 57.6 |
| **Sentences** | 1.0 | 1.9 | 2.6 | 2.5 | 3.4 | 3.0 |

Table 3: Average length of articles, claims, and verdicts in our dataset.

mandatory. If its exclusion does not detract from the strength of the argument, then it's not necessary to include it. In fact, including extra information may actually be detrimental to the overall readability of the verdict (Lombrozo, 2007; Sanna and Schwarz, 2006).

5. Conversely, new claims or arguments not contained in the original verdict could be generated. If these claims support the argument being presented and are either factual or a subjective opinion, then they were kept. Otherwise, they were removed or rewritten.

### 3.6 Dataset Analysis

After the data augmentation process, we obtained ~12 thousand examples (11990 claim-verdict pairs, 1838 written in a general SMP-style and 10152 also comprising an emotional component). Post-editing details can be found in Appendix A.2. In Table 3 we report the average number of words, sentences, and BPE tokens for the articles, the claims and the verdicts of each stylistic version of our dataset.[6] Then, to quantitatively assess the quality of the post-edited data we employed two measures: the *Human-targeted Translation Edit Rate* (HTER; Snover et al., 2006) and the *Repetition Rate* (RR; Bertoldi et al., 2013).

**HTER** measures the minimum edit distance, i.e. the smallest amount of edit operations required, between a machine-generated text and its post-edited version. HTER values greater than $0.4$ account for low-quality generations; in this case, writing a text anew or post-editing would require a similar effort (Turchi et al., 2013). In Table 4 we report the HTER of the post-edited claims and verdicts [7].

---

[6]We employed Spacy (https://spacy.io) for extracting words and sentences, and the sentence-piece tokenizer used in Pegasus (Zhang et al., 2020) for the BPE tokens.

[7]HTER values were averaged over the entire samples under analysis (including the non-post-edited data)

**RR** measures the repetitiveness of a text, by computing the geometric mean of the rate of n-grams occurring more than once in it. A fixed-size sliding window while processing the text ensures that the differences in documents' size do not impact the overall scores. For our analysis, we computed the rate of word n-grams (with n ranging from 1 to 4) with a sliding window of 1000 words. Following previous works (Bertoldi et al., 2013; Tekiroğlu et al., 2020), the RR values reported in this paper range between 0 and 100.

As can be seen in Table 4, the HTER values computed on the claims are very low, always less than 0.1, suggesting good quality machine-generated texts. In particular, the data generated according to a general SMP-style were less post-edited. Machine-generated claims, which comprise also an emotional component, required more post-editing than SMP-style claims, as shown by the higher HTER values.

Moreover, HTER values for the post-edited verdicts are higher than those for the claims. This can be explained by the need to ensure verdicts' truthfulness, by adjusting or removing calls to action, possible model hallucinations or repeated patterns. However, even though HTER values vary across the single emotions, on average they are lower than the 0.4 threshold.

This is corroborated by the RR of the verdicts: a substantial decrease in repetitiveness was obtained after post-editing at the expense of more editing operations. The average RR for the data comprising an emotional component is comparable to the one obtained on the corresponding claims. However, this does not apply to the SMP-style data: in this case, the RR for the claims is more than 2 points lower than that on the verdicts. This can be explained by the tendency of the LLMs employed to produce more recurrent patterns when the instructions are enriched with specific details, such as the emotional state.

In summary, our pipeline facilitated the acqui-

| | | FF | SMP-style | happiness | anger | fear | disgust | sadness | surprise | all emotions |
|---|---|---|---|---|---|---|---|---|---|---|
| | # samples | 1838 | 1838 | 1527 | 1590 | 1805 | 1675 | 1758 | 1797 | 10152 |
| **claims** | **HTER** | - | 0.028 | 0.066 | 0.055 | 0.058 | 0.060 | 0.047 | 0.073 | 0.059 |
| | **RR** generated | - | 1.578 | 4.795 | 4.194 | 5.784 | 5.066 | 6.068 | 5.739 | 3.903 |
| | post-edited | - | 1.501 | 4.803 | 4.206 | 5.800 | 5.089 | 6.149 | 5.692 | 3.945 |
| **verdicts** | **HTER** | - | 0.275 | 0.335 | 0.339 | 0.338 | 0.317 | 0.319 | 0.262 | 0.318 |
| | **RR** generated | - | 6.359 | 6.742 | 7.155 | 7.266 | 7.355 | 6.938 | 6.482 | 6.761 |
| | post-edited | - | 3.872 | 4.292 | 4.128 | 4.250 | 4.149 | 4.476 | 4.168 | 4.200 |

Table 4: For each dataset (column-wise): number of samples, HTER and Repetition Rate (RR) values for both the post-edited claims and verdicts.

sition of a substantial volume of data while simultaneously minimizing the annotators' workload.[8] Additionally, the intervention of the annotator substantially increases the quality of the data as reported in the lowered values of RR.

## 4 Experimental Design

Inspired by the summarization approaches proposed by Atanasova et al. (2020); Kotonya and Toni (2020b), for the automatic generation of personalised verdicts we leveraged an LM pretrained with a summarization objective. To overcome the limitation of the model's fixed input size, we reduced the length of the input articles, by adding an extractive summarization step beforehand (following the best configuration presented in Russo et al., 2023). This extractive-abstractive pipeline was tested on different configurations, both in in-domain and cross-domain settings.[9] The quality of the generated verdicts was assessed with both an automatic and a human evaluation. We present and discuss the results in Section 5.

### 4.1 Extractive Approaches

Under an extractive summarization framing, we defined the task of verdict generation as that of extracting 2-sentence long verdicts from FullFact articles, and 3-sentences long for the SMP and emotional data. Such lengths were decided according to the averaged length of the verdicts, reported in Table 3. We employed SBERT (Reimers and Gurevych, 2019), a BERT-based siamese network used to encode the sentences within an article as well as the claim and to score their similarity via

cosine distance (SBERT-k henceforth, with $k$ denoting the number of sentences). Under our experimental design, the top-k sentences with a latent representation closer to that of the claim would be selected to construct the output verdict.

We used a semantic retrieval approach rather than other common unsupervised methods for extractive summarization, such as LexRank (Erkan and Radev, 2004), since the latter has no visibility into the claim itself. Nonetheless, we tested those approaches in preliminary analyses (reported in Appendix C.1) and verified that the performance was significantly lower than that obtained with SBERT. We will consider SBERT-k as a baseline for the following experiments.

### 4.2 Abstractive Models

We employed PEGASUS (Zhang et al., 2020), a language model pretrained with a summarization objective. In all the experiments, the length of the articles was reduced through extractive summarization with SBERT (Reimers and Gurevych, 2019) in order to fit the maximum input length of the model (i.e. 1024). We opted for SBERT in light of its higher performances with respect to other extractive methods (see Appendix C.1). We explored four different configurations and tested them on all the versions of our dataset, i.e. FullFact, SMP, and emotional version (see Appendix C.2 and C.3 for fine-tuning and decoding details):

- $\text{PEG}_{base}$: Zero-shot experiments with PEGASUS fine-tuned on CNN/Daily Mail,[10] with the goal of summarizing the debunking article.

- $\text{PEG}_{FF}$: Fine-tuning of $\text{PEG}_{base}$ on FF data. A claim and its corresponding debunking article were concatenated and used as input, with the verdict as target.

---

[8]This is corroborated not only by the HTER values constantly lower than 0.4 but also by a supplementary experiment presented in Appendix A.3, which revealed that creating content from scratch takes roughly three times more than post-edit machine-generated data.

[9]In-domain refers to train and test data having the same style, while cross-domain involves data with different styles.

[10]https://huggingface.co/google/pegasus-cnn_dailymail

| Model | VerMouth-test | R1 | R2 | RL | METEOR | BARTScore | BERTScore |
|---|---|---|---|---|---|---|---|
| SBERT-2 | FullFact | **.245** | **.092** | **.180** | **.328** | **-2.898** | **.874** |
| SBERT-3 | SMP-style | .230 | .054 | .149 | .268 | -2.981 | .863 |
| SBERT-3 | emotional style | .228 | .051 | .144 | .251 | -3.091 | .858 |
| $PEG_{base}$ | FullFact | **.223** | **.073** | **.159** | **.291** | **-3.124** | **.856** |
| $PEG_{base}$ | SMP-style | .217 | .045 | .139 | .213 | -3.181 | .852 |
| $PEG_{base}$ | emotional style | .217 | .044 | .141 | .202 | -3.253 | .849 |
| $PEG_{FF}$ | FullFact | **.282** | **.104** | **.213** | **.345** | **-2.824** | **.886** |
| $PEG_{FF}$ | SMP-style | .244 | .058 | .162 | .227 | -3.079 | .873 |
| $PEG_{FF}$ | emotional style | .233 | .052 | .155 | .203 | -3.173 | .867 |
| $PEG_{smp}$ | FullFact | .260 | .084 | .184 | .297 | -3.038 | .883 |
| $PEG_{smp}$ | SMP-style | **.337** | **.127** | **.240** | **.320** | **-2.864** | **.896** |
| $PEG_{smp}$ | emotional style | .323 | .121 | .229 | .301 | -2.918 | .890 |
| $PEG_{emo}$ | FullFact | .246 | .078 | .175 | .286 | -3.084 | .877 |
| $PEG_{emo}$ | SMP-style | .326 | .124 | .233 | .321 | -2.858 | .892 |
| $PEG_{emo}$ | emotional style | **.337** | **.131** | **.234** | **.331** | **-2.810** | **.893** |

Table 5: Results for each configuration, for both the in-domain and cross-domain experiments.

- **$PEG_{smp}$**: Fine-tuning of $PEG_{base}$ on the SMP-style data. Training input data were processed as in the $PEG_{FF}$ configuration.

- **$PEG_{emo}$** Fine-tuning of $PEG_{base}$ on the *emotional* data.[11] Training input data were as in the $PEG_{FF}$ configuration.

## 5 Results

We assessed the potential of our proposed dataset in terms of generation capabilities via both automatic and human evaluation.

### 5.1 Automatic Evaluation

We adopted the following automatic measures:

- **ROUGE** (Recall-Oriented Understudy for Gisting Evaluation; Lin, 2004) measures the overlap between two distinct texts by examining their shared units. We include ROUGE-N (*RN*, N=1,2) and ROUGE-L (*RL*), a modified version that considers the longest common substring (LCS) shared by the two texts.

- **METEOR** (Banerjee and Lavie, 2005) determines the alignment between two texts, by mapping the unigrams in the generated verdict with those in the reference gold verdict, accounting for factors such as stemming, synonyms, and paraphrastic matches.

- **BERTScore** (Zhang et al., 2019) computes token-level semantic similarity between two texts using BERT (Devlin et al., 2019).

- **BARTScore** (Yuan et al., 2021), built upon the BART model (Lewis et al., 2020), frames the evaluation as a text generation task by computing the weighted probability of the generation of a target sequence given a source text.

Table 5 reports the results of all the experiments we carried out. For all metrics, the higher scores were obtained after fine-tuning the model in both in-domain and cross-domain experimental scenarios. Indeed, zero-shot experiments with $PEG_{base}$ resulted in scores even lower than the SBERT baseline. This suggests that summarising the article is not enough by itself to obtain quality verdicts.

Interestingly, the PEGASUS models fine-tuned on the SMP-style and emotional style samples appear to generalise better. In fact, when tested against the other test subsets, they have a similar overall performance and a smaller decrease in cross-domain settings compared to $PEG_{FF}$.

### 5.2 Human Evaluation

We adapted the methodology proposed by He et al. (2023) to our scenario: three participants were asked to analyse 180 randomly sampled items; each item comprises the claim and three verdicts produced by $PEG_{FF}$, $PEG_{smp}$ and $PEG_{emo}$ over that claim, compounding to 60 claims for each stylistic configuration present in VerMouth.

---

[11]In order to fairly compare the models' performance, we carried out a stratified subsampling of the emotional data, so that the size of the train and evaluation sets was equal across all the different configurations tested.

We asked to evaluate the model-generated verdicts by answering the following question:

*Consider a social media post, which response is better when countering the possible misinformation within the post (the claim)? Rank the following responses from the most effective (1) to the least effective(3). Ties are allowed.*

After collecting the responses, we run a brief interview to understand the main elements that drove the annotators' decisions. These interviews highlighted some crucial aspects: (i) verdicts comprising too much data and information induced a negative perception of their effectiveness (overkill backfire effect); (ii) verbose explanations are generally not appreciated; (iii) there was a positive appreciation for the empathetic component in the response, however (iv) "over-empathising" was negatively perceived.

Table 6 shows how $PEG_{FF}$ is highly preferred for in-domain cases, possibly because it avoids (i) excessively long verdicts and (ii) the stylistic/empathetic discrepancy between a journalistic claim from FF and other systems' output with a more SMP-like style. Still, $PEG_{FF}$ performs the worst in cross-domain settings. Conversely, $PEG_{smp}$ and $PEG_{emo}$ are somewhat more stable (consistently with the automatic evaluation). In general, style and emotions in the verdict have a greater impact if the starting claim has style and emotions. Users reported that empathy mitigates the length effect. From a manual analysis, $PEG_{smp}$ shows the ability to provide slightly empathetic responses, so it sometimes ended up being preferred for its empathetic (but not overly so) responses.

To sum up: for social claims, which resemble those found online, social verdicts are widely preferred to FullFact journalistic claims.

|  | FF | SMP | EM |
|---|---|---|---|
| $PEG_{FF}$ | 1.55 | 2.08 | 2.00 |
| $PEG_{smp}$ | 1.93 | 1.97 | 1.75 |
| $PEG_{emo}$ | 1.90 | 1.92 | 1.80 |

Table 6: Average rankings obtained via human evaluation. The ranks range from 1 (most effective) to 3 (least effective). The best results are highlighted in blue.

## 6 Conclusion

Producing a verdict, i.e. a factual explanation for a claim's veracity and doing so in a constructive and engaging manner is a very demanding task. On social media platforms, this is usually done by ordinary users, rather than professional fact-checkers. In this context, automated fact-checking can be very beneficial. Still, to fine-tune and/or evaluate NLG models, high-quality datasets are needed. To address the lack of large-scale and general-domain SMP-style resources (grounded in trustworthy fact-checking articles) we created VerMouth, a novel dataset for the automatic generation of personalised explanations. The provided resource is built upon debunking articles from a popular fact-checking website, whose style has been altered via a collaborative human-machine strategy to fit realistic scenarios such as social-media interactions and to account for emotional factors.

## Limitations

There are some known limitations of the work presented in this paper. First, the resource is limited to only English language only; nonetheless, the author-reviewer approach we adopted for data collection is language-agnostic and can be transferred as-is to other languages, assuming the availability of (i) a seed set of *<article,claim,verdict>* triples for (or translated in) the desired target language, and (ii) an instruction based LLM for the desired language. Furthermore, this dataset is limited in the sense that it only covers a particular style of language most commonly seen on specific Social Media Platforms – short and informal posts (such as those typically found on Twitter and Facebook), rather than longer or more formal posts (which may be more typical on sites such as Reddit or on internet forums). We leave efforts to tackle such limitations to future iterations of this work.

## Ethics Statement

The debate on the promise and perils of Artificial Intelligence, in light of the advancements enabled by LLM-based technologies, is ongoing and extremely polarising. A common concern across the community is the potential undermining of democratic processes when such technologies are coupled with social media and used with malicious/destabilising intent. With this work, we provide a resource aiming at countering such nefarious dynamics while integrating the capabilities of LLMs for social good.

# Acknowledgements

This work was partly supported by the AI4TRUST project - AI-based-technologies for trustworthy solutions against disinformation (ID: 101070190).

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

## A  Data Augmentation Details

### A.1  Prompt and Model Selection

The first step when it comes to effectively leveraging LLMs for one's specific use case is prompt engineering. In our case, a carefully crafted prompt allows LLMs to produce quality claims and verdicts systematically, minimising the amount of post-editing required. Since ChatGPT's API was not yet available to the public when we began our experiments, the initial prompt testing was performed using GPT3.

Initial tests focused on finding the optimal prompt and parameters for our specific use case. The parameters we tested were the *temperature $T$* and the cumulative probability $p$ for *nucleus sampling (Top-P)*.

The $T$ hyperparameter ranges between 0 and 1 and controls the amount of randomness used when sampling: a value of 0 corresponds to a deterministic output (i.e. picking exclusively the top-probability token from the vocabulary); conversely, a value of 1 provides maximum output diversity. Finding a balance between not being overly deterministic while remaining coherent was the goal when testing different temperature values - 0.7 and 1 were used when performing these initial tests.

The *Top-P* parameter also ranges between 0 and 1 and determines how much of the probability distribution of words is considered during generation. It was necessary to find a value which was not overly deterministic and that avoid using very rare words that may reduce coherence. During these initial tests, we tried using the default value of 0.5 as well as a Top-P value of 1, which includes all words in the probability distribution. We also tested using a Top-P of 0.9.

Determining the "optimal" prompt and parameters can be challenging because what makes a "good" personalised claim or verdict is subjective. Several factors were taken into account when selecting our prompts and parameters:

1. **Generalisability**: Do they perform well on a variety of claims, or do they only work well on specific ones (ie: it produces quality output for Covid-related claims, but struggles with claims about Brexit)?

2. **Variability**: Do the generated claims and verdicts vary between one another, or do they all follow similar patterns?

3. **Originality**: Do the generated claims and verdicts resemble too much the original? Do they contain the original claim or verdict verbatim?

4. **Coherence**: Do the generated claims and verdicts make sense? Are they coherent? Are they saying what the original claims and verdicts are, or do they instead say something unrelated?

5. **Amount of Post-Editing**: On average, how much post-editing is required for each of the generated claims and verdicts? What proportion of these claims and verdicts requires any post-editing at all?

Eventually, we opted for the following parameters: a temperature of 1 and a Top-P of 0.9. These parameters were used with OpenAI's Davinci model during initial tests. No changes were made when we switched to ChatGPT after their public API was released.

### A.2  Post-Editing

The post-editing of the data and the prompt evaluation were carried out by two annotators either native English speakers or fluent in English. The time needed for post-editing was heavily dependent on the type of data (claim versus verdict) and on the configuration (SMP-style versus emotional style). On average, the annotators were able to process 250 SMP-style claims and 200 emotional claims per hour [12]. For the emotional data, the time required to post-edit varied greatly depending on the emotion. Since verdicts are usually longer than claims and much more constrained, the post-editing process was much longer: on average, 150 SMP-style verdicts and 70 emotional verdicts were able to be post-edited per hour.

Not every claim and verdict required post-editing. Only ~19% of the generated SMP-style claims and ~68% of the emotional style claims were post-edited. Keep in mind that some of the generated emotional claims were discarded if a specific emotion and the content of the claim were mismatched, as this resulted in forced and unnatural combinations. Figure 2 displays the distribution of post-edited, non-post-edited, and discarded claims. Fear and surprise were the emotions with the least

---

[12]The annotators were timed on several occasions during post-editing (always sessions of 20 minutes to make fair comparisons); the results reported in the paper are the average of the claim/hour amount measured in each session.

amount of discarded data, but they also required the most post-editing.

As mentioned before, post-editing verdicts were a much longer process. Since verdicts are subject to stricter standards of quality (because they must be truthful and polite, for example) and are much longer on average, many more of them required post-editing. Figure 3 shows this disparity: for some emotions such as disgust and fear, there were fewer than 10 verdicts which did not require at least minimal post-editing. SMP-style verdicts also required post-editing at a much higher rate than SMP-style claims, although less than emotional style verdicts. In total, there were 1838 SMP-style verdicts and 2609 emotional style verdicts, resulting in post-editing rates of 91.4% and 96.3% respectively.

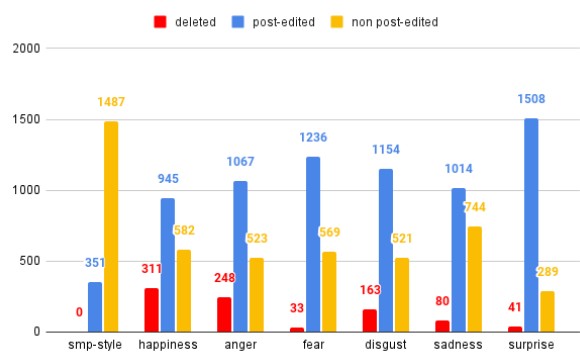

Figure 2: Graph representing the number of ChatGPT generated claims that were deleted, post-edited, or not post-edited

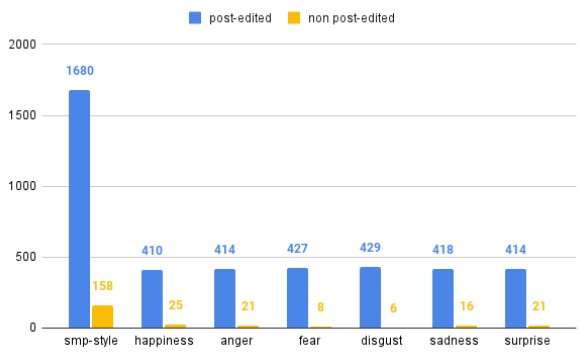

Figure 3: Graph representing the number of ChatGPT generated verdicts that were post-edited versus those which were not post-edited

## A.3 Timing benefits of post-editing

We carried out an extra experiment to assess whether post-editing machine-generated data is more effective in terms of time than writing new data from scratch. To this end, we provided one of the annotators with 60 claims and asked to write from scratch new tweets, 30 in SMP-style and 30 emotional tweets. In both cases, it took the annotator (expert in the field) around 23 minutes to create 30 new tweets (thus, roughly 80 claims per hour as compared to the 250 SMP-style and 200 emotional tweets obtained with our pipeline). If in the creation of claims the time differences are considerable, we assume that this also applies to verdicts, which is a task that requires more constraints.

## B Detailed Guidelines

### B.1 Claim Guidelines

1. **Generated claims copying original claims verbatim**: Sometimes the wording of the original claim is copied verbatim in the generated claim. These should be rewritten to avoid resembling the original dataset. Determining whether a generated claim resembles the original claim "too much" can be subjective, so discretion must be used.

2. **Reoccurring Patterns**: Since the SMP-style claims have a lot more freedom to decide what sort of tone to adopt, they are much more diverse. With emotional style claims, the emotional component is an extra constraint which is applied during generation. This means that there are often reoccurring patterns which appear in the resulting generated claims: *"I'm livid!"*, *"'So sad to hear that X"*, *"Disgusting!"*, etc. If a pattern can be removed while preserving the overall emotional intent, then it is better to remove it entirely. Conversely, if removing a pattern also removes any *"emotion"* from the generated claim, then rewriting is preferred. In rare cases, the pattern can be kept, keeping in mind that too many occurrences may result in degraded performance during training.

3. **Hashtags**: Due to the prompt used, hashtags often appear in the generated claims. We noted two different phenomena which may occur and which require post-editing:

   (a) **Debunking hashtags**: There are some occurrences where a hashtag debunks a claim or works against the claim's intent. If a claim is about how vaccines are not effective, having the hashtag "#VaccinesSaveLives" is not appropriate. These hashtags

can either be removed or edited to match the original intent.

(b) **Unnecessary hashtags**: There are some hashtags which are so vague that they diminish the overall quality of the claim (such as *"#miracle"*, *"#goodjob"*, etc.). In emotional claims specifically, the emotion given in the prompt is turned into a hashtag (such as *"#happy"* or *"#sad"*). Any hashtags which fit these criteria are to be removed.

4. **Generated claims debunking original claims**: There are some instances where the generated claim actually *debunks* the original claim. These must be changed to reflect the intent behind the original claim. For example: if the original claim says that *"wearing masks causes dementia and hypoxia"* and the generated claim says *"False info alert: wearing a mask doesn't cause dementia and hypoxia"*, it should be rewritten to match the original claim.

5. **Dates and places**: The original claims contained many references to dates and places. These could be vague references (*"last year"*, *"in our nation"*, etc.) or specific references (*"21 July 2021"*, *"in England"*, etc.). Any dates and places in the generated claims should match the original claim's level of specificity.

6. **Hallucinations**: Since claims do not necessarily need to be *true*, hallucinations can often be beneficial. Consider an example where the original claim says *"almost 300 people have died from the vaccine"*, but the generated claim contains a hallucination which states that *"291 people have died from the vaccine"* - this number is more specific, and this may give off the impression that the person knows what they are talking about.

7. **General formatting issues**: While rare, there are cases in which grammatical errors, typos, malformed hashtags (such as *"#endrape culture"*) or other formatting issues occur in generated claims. These should simply be corrected.

## B.2   Verdict Guidelines

1. **Reoccurring Patterns**: As with generated claims, there are often patterns which occur often in generated verdicts. These should be removed or rewritten. Some examples of common patterns include *"thank you for X"*, *"I under-stand that you feel X"*, *"Let's continue to follow the recommended guidelines"*, etc.

2. **Calls to action**: As stated before, a *"call to action"* is a phrase or sentence which, as the name implies, calls upon the reader of the verdict to take action in some way. For example:

*"I understand your frustration, and while the proportion of BME students at Oxbridge has actually increased, I agree that more needs to be done to address the lack of diversity from disadvantaged areas. **It's important that we continue examining the root causes of this inequality and work towards equal opportunities for all.** #diversitymatters #educationforall"*

To avoid overtly political or polarising verdicts, many considerations need to be kept in mind when a call to action in a generated verdict is encountered.

(a) **Is the call to action well-integrated into the verdict?** - If a call to action does not make a meaningful contribution to the overall quality of the verdict, then it will be removed. An example of a poorly-integrated call to action is *"let's focus on promoting peaceful and respectful discourse."* This call to action is broad and vague and should be removed.

(b) **Is the call to action political or polarising?** - One must determine whether or not the call to action is actually political or polarising. With certain topics, it is simply impossible to avoid having a call to action which contains political elements (such as a claim about a politician, or a new law). We decided upon two possible approaches one can take when post-editing political calls to action.

The *common sense approach* (or the *"reasonable person" approach*) is employed for a call to action expressing a political opinion on which the most agree (such as *"demanding transparency and accountability from our government"*). This call to action can be kept.

The *empathetic approach* is employed when dealing with opinions or thoughts that simply have no "correct" answers, but rely on one's own beliefs. It was decided that the call to action should be changed to empathise with the claim writer's beliefs

without agreeing or disagreeing with them. For example, if a claim expresses pro-life opinions, and a call to action such as *"it's important to acknowledge the magnitude of lives affected by this issue"* exists, changing it to *"it's important to acknowledge the magnitude of lives affected by this issue **no matter what you believe**"* is empathetic, but also explicitly avoids picking one side or the other in a polarising situation like this.

(c) **How strong is a call to action, and who is the focus?** - Different actions require different amounts of effort. If a call to action asks for too much from someone, then it should be changed. Asking someone, for example, to *"advocate for stricter testing protocols"* may be asking too much of them, but asking them to *"hope that stricter testing protocols are implemented"* is not. If a call to action is too strong, then it can either be *weakened* or *neutralised*. Weakening a call to action involves changing what is expected of the reader of the verdict: rather than *"we must personally take action to end child poverty immediately"*, one can post-edit the call to action to say *"let's try and do our part together to hopefully end child poverty one day"*.

Neutralising a call to action takes the focus off the reader entirely. This involves either putting the onus on someone else who may be more capable of solving the issue or not demanding action from anyone at all. Thus, *"**we** must continue to keep an eye out for potential side effects of the vaccines"* can be rewritten to *"**the experts** must continue to keep an eye out for potential side effects of the vaccines"*.

An example of not demanding action from anyone at all is: *"we must continue to advocate for those struggling to make ends meet"*. It can be post-edited as *"compassion and understanding for those struggling to make ends meet is crucial"*.

3. **Pronouns and Grammatical Personhood**: As the original verdicts sometimes contain first-person plural pronouns (*"we have contacted them for more clarification"*), and at other times contained first-person singular pronouns (*"I understand your frustration"*), there are incon-sistencies regarding "who" is writing the verdict. The assumption one should take when post-editing is that each verdict is written by a single person. Therefore, if first-person plural pronouns are encountered, they should be changed to first-person singular pronouns.

One exception exists: when the reader and writer of the verdict are grouped together, then first-person plural pronouns can be kept: *"surely **we** can all agree that this is a serious issue"*.

4. **Confirmations**: Sometimes a claim is fully or partially correct, and the original FullFact verdict notes this with a simple *"correct"*, or *"this is right, but.."*, but the generated verdict does not.

In this case, adding a quick confirmation such as *"yes, you're right, but.."* or *"absolutely, it's a serious issue"* can be done as long as it does not reduce the overall readability of the verdict.

5. **Missing information**: Sometimes the generated verdicts do not include information from the original verdicts. This can make the generated verdict easier to read without reducing its persuasiveness. If missing information negatively impacts how effective a verdict is, then it should be added.

6. **New claims**: Conversely, there are cases in which the generated verdicts actually include information which is not contained in the original verdict, but which is either objectively true or is a subjective statement. If the claims made in the generated verdict are provably false, then removing them is necessary. If they are provably true, or if they are a subjective statement or opinion which can not be concretely proven true or false, they can be kept or removed at the post-editors discretion.

7. **General formatting issues**: As with generated claims, general formatting issues such as grammatical errors, typos, malformed hashtags, etc. should be corrected.

## C   Experimental Details

### C.1   Extractive Summarization Methods

Besides SBERT, we also considered other extractive summarization methodologies, i.e. **Lead-k** which extract the first $k$ sentences from the article, and **LexRank** (Erkan and Radev, 2004), a graph-based unsupervised methodology which ranks the

|          | R1   | R2   | RL   | METEOR | BARTScore | BERTScore |
|----------|------|------|------|--------|-----------|-----------|
| **Lead-2**   | 0.21 | 0.06 | 0.15 | 0.26   | -3.23     | 0.86      |
| **Lexrank**  | 0.18 | 0.06 | 0.12 | 0.31   | -2.89     | 0.86      |
| **SBERT**    | **0.25** | **0.09** | **0.18** | **0.33** | **-2.90** | **0.87** |

Table 7: Results for the extractive summarization methodologies on FullFact dataset.

| **Gold Verdict** | | The network has been attacked, but the idea that it was Islamic State appears to be based on speculation from a single, unnamed student. |
|---|---|---|
| **FullFact** | claim | Islamic State has hacked a major computer network used by universities. |
| | model | This is not the case. The network provider, Jisc, has told us that "the identity of the perpetrators remains subject to an ongoing criminal investigation" |
| **SMP-style** | claim | Just heard Islamic State has breached a large computer network utilized by universities! Super scary & unnerving–hopefully authorities can help resolve the issue ASAP. #Ishacked |
| | model | It's definitely alarming to hear about the breach of a university's computer network. However, it's important to note that there is currently no evidence linking Islamic State to the attack. Hopefully authorities can resolve the issue quickly. |
| **Emotional** | claim | As a student, I am infuriated by the news that ISIS has hacked into a major university computer network. This is a violation of our education and a threat to our safety. #stopISIS |
| | model | It's important to note that the news about ISIS hacking into a major university computer network has not been confirmed by any official source. Nonetheless, it's understandable to be concerned about the potential impact this may have on our education and safety. #stopISIS |

Table 8: Examples of generated verdicts for each fine-tuning configuration tested in-domain.

sentences of a document based on their importance by means of eigenvector centrality. We tested these approaches by extracting two sentence-long summaries from Fullfact articles. Subsequently, these summaries were evaluated against the gold verdicts. The results of this evaluation are shown in Table 7. SBERT outperforms both Lead-2 and LexRank for all the metrics employed.

### C.2 Fine-Tuning Configuration

When fine-tuning, $PEG_{base}$ was trained for 5 epochs with a batch size of 4 and a random seed set to 2022. To this end, we employed the Hugging-face Trainer [13] using the default hyperparameter settings, with the exception of the Learning Rate values and the optimisation method. Instead, we used the Adafactor stochastic optimisation method (Shazeer and Stern, 2018) and a Learning Rate value of 3e-05. The training was performed on a single Tesla V100 GPU, while the testing was performed on a single Quadro RTX A5000 GPU. The checkpoint with minimum *evaluation loss* was employed for testing.

### C.3 Decoding Configuration

At inference time, we employed *nucleus sampling* decoding strategy, setting the probability at 0.9, and repetition penalty, set at 2.0, for the verdict generation.

### D Examples of Generated Verdicts

In Table 8 we report examples of verdicts generated with PEGASUS model (Zhang et al., 2020) fine-tuned on the three different stylistic versions of our dataset, i.e. FullFact, SMP-style and emotional style. In particular, we report the generations obtained in the in-domain experiments.

---

[13]https://huggingface.co/docs/transformers/main_classes/trainer