# OpenReview forum: "Countering Misinformation via Emotional Response Generation"
_EMNLP/2023/Conference — EMNLP 2023 Main_

### Official Review · Reviewer_NTyP · 2023-08-01

**Soundness:** 4

**Excitement:**

4: Strong: This paper deepens the understanding of some phenomenon or lowers the barriers to an existing research direction.

**Paper Topic And Main Contributions:**

This paper focuses on debunking misinformation with the discourse style and proper emotional content of social media. They contribute a dataset of 12k claim-response pairs generated via an LLM with human post-editing. They show improvements over baselines that are not in the style of social media with automatic summarization-based evaluations and human evaluations.

**Questions For The Authors:**

A. How were the annotators recruited and how many of them were there? I'm interested in those who did post-editing and also those doing evaluation. Please provide a line number if I simply missed this information.

B. Were the articles used for evaluation also included in the SMP-style and emotional datasets? Or are these unseen articles?

C. Why was the "Make it personal" sentence included in prompts for the emotional style in claim generation?

**Reasons To Accept:**

The task is well-motivated in that effective misinformation debunking must address the norms of social media. The dataset seems useful for future work, and the evaluation (particularly the human evaluations) show the benefit of the dataset toward this task.


**Reasons To Reject:**

* There is a lack of sufficient detail on recruitment of the annotators. (See question).
* I understand hallucinations in claim generation are expected, but the "as someone who has lost a loved one" sentence in the example claim makes me wonder about the impact of the "make it personal" prompt to author LLM for the emotional claims. Not everything on social media is personal even if emotional (people often just give impassioned statements of opinion) and could potentially lead to the model over-generating personal stories in claims. Or perhaps this is just a matter of that example claim. (See question.)

**Reproducibility:**

4: Could mostly reproduce the results, but there may be some variation because of sample variance or minor variations in their interpretation of the protocol or method.

**Reviewer Confidence:**

4: Quite sure. I tried to check the important points carefully. It's unlikely, though conceivable, that I missed something that should affect my ratings.

**Typos Grammar Style And Presentation Improvements:**

* Giving a ballpark range of the "limited size" of other datasets (last para of section 2) would be helpful
* Line 41: "fake news"?
* I found the RR explanation confusing. What is "n-grams' rate" exactly? Is it the rate of repeated n-grams?
* Line 472: don't capitalize "The"
* Table 6 took me a while to understand. It should be clearer (in the caption) that columns are different datasets (starting claims) and rows are models. I also would have expected the blue highlights for best results to be best per dataset instead of best per model, since that would give more expected information when comparing models for performance across datasets (which I think is the purpose) vs comparing datasets to see the areas in which models perform the best.

---

> ### Author Rebuttal · Authors · 2023-08-28
>
> Dear reviewer,
>
> We thank you for your feedback. We will make sure to integrate the comments and suggestions you provided into the final version of our paper if accepted. Hereafter, we will try to answer the questions and doubts you raised in your review.
>
> > *A. How were the annotators recruited and how many of them were there? I'm interested in those who did post-editing and also those* doing evaluation. Please provide a line number if I simply missed this information.
>
> We understand that the information regarding the annotators could have been enriched and we apologise for the lack of details. Specifically, the data generated with the LLMs were post-edited by two annotators: one last-year master's student and a Ph.D. student. Both of them are included as authors of the present work (in addition, the master's student got their university credit and the PhD has a salary). Both annotators were extensively trained on the data and the topic of misinformation and automated fact-checking, as well as they had good experience with the communication style employed on the main social media platforms. In particular, before starting to work on the data they were properly trained for the task, by (i) providing relevant literature, (ii) walking them through the complete guidelines, (iii) providing and discussing several claim and verdict examples, and (iv) answering possible doubts. Moreover, weekly meetings were organized throughout the whole annotation campaign to discuss possible problems and doubts about specific post-editing instances. We will include these details in the additional page provided for the final version of the paper upon acceptance.
>
> > *B. Were the articles used for evaluation also included in the SMP-style and emotional datasets? Or are these unseen articles?*
>
> For the human evaluation, we created subsamples of data from the verdicts generated by our models (in the various configurations) while inferencing from the test set. The test set comprises data from FullFact (*journalistic style*) and from VerMouth (*SMP-style* and *emotional-style* data).
>
> >*C. Why was the "Make it personal" sentence included in prompts for the emotional style in claim generation?*
>
> We used the "Make it personal" sentence in order to obtain more realistic claims, but not all generated claims reported a personal story and as the reviewer correctly pointed out "it is just a matter of that example claim". We will clarify this aspect in the final version of the paper if it is accepted.

---

### Official Review · Reviewer_LBT1 · 2023-08-07

**Soundness:** 3

**Excitement:**

3: Ambivalent: It has merits (e.g., it reports state-of-the-art results, the idea is nice), but there are key weaknesses (e.g., it describes incremental work), and it can significantly benefit from another round of revision. However, I won't object to accepting it if my co-reviewers champion it.

**Missing References:**

Connected to studies on misinformation and fact-checking on Twitter, some papers look at community-oriented fact-checking with Birdwatch (now called ‘community notes’). While these approaches do not use automated fact-checking, these datasets model more naturalistic fact-checking behavior on social media sites and can be informative about the ideal characteristics of a good verdict. Some papers in this vein are:

Pröllochs, Nicolas. "Community-based fact-checking on Twitter’s Birdwatch platform." Proceedings of the International AAAI Conference on Web and Social Media. Vol. 16. 2022.

Allen, Jennifer, Cameron Martel, and David G. Rand. "Birds of a feather don’t fact-check each other: Partisanship and the evaluation of news in Twitter’s Birdwatch crowdsourced fact-checking program." Proceedings of the 2022 CHI Conference on Human Factors in Computing Systems. 2022.


**Paper Topic And Main Contributions:**

The paper contributes a dataset of claim-response pairs to be used in developing and evaluating automated methods for countering misinformation through social correction. The presented dataset builds on a collection of fact-checked claims and professional fact-checkers’ verdicts and is extended through the transfer of the verdicts from the journalistic style of professionals to the style typically encountered on social media platforms, among other aspects richer in basic emotional content. The authors present this style transfer as their main contribution, as they claim that claim-verdict pairs in a typical social media style is crucial for developing automated methods to effectively support social correction efforts, ultimately helping to counter misinformation on social media platforms.

**Questions For The Authors:**

Question A: How exactly was the data “obtained from the FULLFACT-Website”? Particularly, from the description in the paper, there is no way of discerning whether the collected data represents all the fact-checks available from the source at the time of data collection or whether some selection criteria were explicitly or implicitly used. Connected, an overview of the number of collected instances per topic would be interesting, especially given that different topics are already mentioned in the paper. This could be an important piece of information for potential re-users of the dataset.

Question B: Who were the annotators that post-edited the claims and verdicts, i.e., how were they recruited and selected? How were they remunerated? Were the same annotators responsible for the quality evaluation of different LLM prompt setups, for the post-editing of claims and verdicts, and the human evaluation study, or were these different sets of annotators?

QuestionC: How much did ChatGPT hallucinate for the claims and verdicts? As the paper states, hallucinations are not very problematic for claims (unless it completely changes the topic), but it is for verdicts. How prevalent were these? And how can we conclude that this pipeline reduces the annotation burden instead of increasing it without a comparative experiment?


**Reasons To Accept:**

The authors make a very convincing claim that their proposed style transfer from journalistic fact-checks to social media-like claims and verdicts helps develop and evaluate automated methods to support social correction efforts. Their own Human Evaluation study shows that automated methods fine-tuned on their contributed dataset generate verdicts that are preferred over verdicts generated from models fine-tuned on journalistic claims and verdicts, with additional interviews with their study’s participants surfacing that the empathetic component in their model’s generation was positively appreciated. The strong points of the paper are:
- The authors create a human-AI collaborative pipeline for data augmentation that, in principle, reduces manual annotation effort
- The authors also conduct convincing quantitative analyses of the generated datasets.


**Reasons To Reject:**

My main concern with the paper and the contributed dataset is only superficially addressed by the authors - the issue of LLM hallucinations.

While the authors convincingly reason that hallucinations in the generated claims are not problematic and in some cases even improving the claim’s quality, this does not hold for the verdicts. While the authors also concede this - albeit only in the discussion of generated verdicts and the Verdict Guidelines included in the appendix - it is unclear how exactly they addressed this issue to ensure that no hallucinated false claims ended up in the verdicts of their datasets. Given the potentially devastating effects of sharing a dataset of claim-verdict pairs as a resource to develop and evaluate methods for the automated generation of factually correct verdicts to counter misinformation through social correction, I would expect the authors to provide more info on the prevalence of the problem (how many generated verdicts contained claims that were provably false and had thus to be removed, as laid out to the annotators in the Verdict Guidelines) and on the measures they took to ensure that their contributed dataset is free of hallucinated false claims, as the task of detecting potentially false claims, fact-checking them, and deleting them if false, seems to be rather difficult for single human annotators that post-edited at a speed of 150 (for SMP; 70 for emotional) verdicts per hour.

Next, and related to the first point, is the hybrid data augmentation strategy, The paper claims that using ChatGPT to reword the claims and verdicts minimizes the annotators’ effort and then provides detailed guidelines that the annotators can check for post-editing in Appendix B. Tha main question is, does this really reduce annotators’ efforts? Generally, section 3.2 is missing details of the procedure, whether it saved the annotators time as well as who the annotators were, etc

Finally, and to some extent, a minor point, but social media style is quite broad. The prompts and the resultant style indicate that the paper wants to investigate Twitter-style claims and verdicts. I would question to what extent this technique is applicable to other social media platforms like Facebook or Reddit.


**Reproducibility:**

2: Would be hard pressed to reproduce the results. The contribution depends on data that are simply not available outside the author's institution or consortium; not enough details are provided.

**Reviewer Confidence:**

4: Quite sure. I tried to check the important points carefully. It's unlikely, though conceivable, that I missed something that should affect my ratings.

**Typos Grammar Style And Presentation Improvements:**

I found the paper mostly well-written and easy to follow, except for the organization of the experiments. It would be good to make clear that Section 4 deals with training and testing models for the verdicts alone. Experiments can also include human evaluations and post-editing, so I found it a bit abrupt that Section 4 only talked about automated experiments. Other than that (L stands for lines in the paper):

L505 introduces ROUGE-L as R-L when abbreviated. Table 5 then uses RL as the abbreviation.

The caption of Figure 1 introduces the <Article, Claim, Verdict> triplets, which are then formatted (and spelled) as <article,claim,verdict> triples in L607.

L948 introduces p as the hyperparameter representing the cumulative probability for nucleus sampling, which then seems to be picked up again only as the Top-P parameter in L960, L967, L969, and L997.

L1024 refers to Figure 2 as Image 2, and L1034 to Figure 3 as Image 3.

---

> ### Author Rebuttal · Authors · 2023-08-28
>
> Dear reviewer,
>
> We really appreciate your detailed review of our paper. While preparing the final version of this paper, if it is accepted, we will improve it by integrating your suggestions and fixing the highlighted typos. Moreover, as already mentioned in the paper, we will make publicly available the resource as well as the accompanying code for reproducibility purposes. Hereafter, we answer the questions raised in your review hoping to clarify your doubts and concerns as comprehensively as possible.
>
> We agree that hallucinations in the verdict are a key issue, for this reason, manual post-editing is of paramount importance. Moreover, it is important to highlight that we prompt the LLM to rewrite a gold verdict and not to write a debunking from scratch. For this reason, the main task of the annotators was to check whether there were discrepancies between the gold and the generated verdicts, and, in case, to correct them. We took for granted that the gold verdicts are trustworthy (as they were manually written by professional fact-checkers), thus we are sure that a new verdict that differs only in style but not in content is trustworthy too. This also explains why the annotators managed to process a high number of data per hour (they did not need to read the whole debunking article to check if assertions in the generated verdict were supported, but only to read the -much shorter- gold verdict).
>
> We did not keep track of the number of hallucinations in the data. Hallucinations were present in claims, but it was not a very common occurrence. As for the verdicts, there were even rarer occasions where it would come up with its own justifications for a specific fact, but it was mostly grounded in the original verdict  (in fact for verdicts we asked to “rephrase” rather than “write”, see Table 2).
>
> There are two main reasons to state that our data augmentation pipeline reduces the annotator's effort:
>
>
> - The HTER values reported in the paper are always much lower than 0.4, meaning that it is easier to post-edit rather than write a tweet from scratch (Turchi et al., 2013).
>
> - We carried out an extra experiment (not reported in the paper, but it will be included in the final version upon acceptance) which proved that human effort is substantially reduced also in terms of time. We provided one of the annotators with 60 claims and asked them to write from scratch new tweets, 30 in SMP-style and 30 emotional tweets. In both cases, it took the annotator (expert in the field) around 23 minutes to create 30 new tweets (thus, roughly 80 claims per hour as compared to the 250 SMP-style and 200 emotional tweets obtained with our pipeline). If in the creation of claims the time differences are considerable, we assume that this also applies to verdicts, a task that requires more constraints.
>
> We believe that this technique could be applicable also to other social media communication styles (such as Facebook and Reddit) after a proper study of the most effective prompt for the scenario under analysis.
>
> As regards the FullFact data, we scraped all the fact-checking instances present on the website within the date range indicated in the paper. Further details about the dataset (including the topic distribution) will be provided in the final version of the paper, and we thank the reviewer for pointing this out.
>
> We understand that the information regarding the annotators could have been enriched and we apologise for the lack of details. Specifically, the data generated with the LLMs were post-edited by two annotators: one last-year master's student and a Ph.D. student. Both of them are included as authors of the present work (in addition, the master's student got their university credit and the PhD has a salary). Both annotators were extensively trained on the data and the topic of misinformation and automated fact-checking, as well as they had good experience with the communication style employed on the main social media platforms. In particular, before starting to work on the data they were properly trained for the task, by (i) providing relevant literature, (ii) walking them through the complete guidelines, (iii) providing and discussing several claim and verdict examples, and (iv) answering possible doubts. Moreover, weekly meetings were organized throughout the whole annotation campaign to discuss possible problems and doubts about specific post-editing instances. We will include these details in the additional page provided for the final version of the paper upon acceptance.
>
> Finally, we really appreciated the suggestion of adding a reference to papers related to community-oriented fact-checking with Birdwatch. We will make sure to properly include them in the final version of our paper.

---

### Official Review · Reviewer_Ysmd · 2023-08-11

**Soundness:** 3

**Excitement:**

3: Ambivalent: It has merits (e.g., it reports state-of-the-art results, the idea is nice), but there are key weaknesses (e.g., it describes incremental work), and it can significantly benefit from another round of revision. However, I won't object to accepting it if my co-reviewers champion it.

**Paper Topic And Main Contributions:**

The paper's main contribution is the claim verification dataset generated using data from the Full-Fact website (FF website). The authors have extensively used LLMs for the purpose of the generation of synthetic samples which are moreover emotion guided. The paper has done analysis with different algorithms to check if the emotion-guided dataset helped in getting clearcut summaries. The results were evaluated with various algorithms with various metrics used in the related works.

**Questions For The Authors:**

Are those 180 samples the same for all 3 human annotators? Is there a plan to release those samples when you will be releasing the dataset?

**Reasons To Accept:**

The paper has solid technical detail as well on the various aspects of augmentation. For example, the authors have pointed out various cases in sections 3.4 and 3.5. I liked the details presented in the paper.

The paper also proves that the dataset is not just another dataset but an important dataset by showing results with the dataset with various algorithms.

The paper builds up gradually on technical details which makes it easy to understand and technical details aren't missed much.

On top of the results with various algorithms, the authors have presented human evaluation as well.

**Reasons To Reject:**

The paper has some details that should be in the main body instead of the appendix.

For example, the authors have mentioned that "Initial tests focused on finding the optimal prompt and parameters for our specific use case". This was done with GPT3 and wasn't changed while using ChatGPT. Any interesting findings that you have documented?

The random 180 samples were chosen for human evaluation using 3 evaluators. Are those 180 samples the same for all 3 annotators? Is there a plan to release those samples when you will be releasing the dataset?

**Reproducibility:**

2: Would be hard pressed to reproduce the results. The contribution depends on data that are simply not available outside the author's institution or consortium; not enough details are provided.

**Reviewer Confidence:**

4: Quite sure. I tried to check the important points carefully. It's unlikely, though conceivable, that I missed something that should affect my ratings.

---

> ### Author Rebuttal · Authors · 2023-08-28
>
> Dear reviewer,
>
> We thank you for your insightful feedback. We will make sure to integrate your comments and suggestions into the additional page provided for the final version of the paper upon acceptance. Moreover, as already mentioned in the paper, upon acceptance we will make publicly available the resource as well as the accompanying code for reproducibility purposes. Hereafter, we will try to answer the questions and doubts you raised in your review.
>
> The decision to move from GPT3 to ChatGPT was taken after some exploratory experiments meant to assess the quality of the generated outputs given the same prompt. To this end, starting from a subset of 50 FullFact claims, SMP-style and emotional claims were generated with both GPT3 and ChatGPT. One of the annotators was required to evaluate (on a scale from 0 to 3) the generated data according to two criteria: post-editing effort and quality of the generation. We opted for the model that showed less post-editing effort and higher quality, namely ChatGPT. In terms of quality, we noticed that ChatGPT:
> - **expands the claims in a more interesting way** (example below)
>
>     **Original claim [emotion: anger]**: "On duty police officers participated in the London Pride parade."
>
>     **GPT3**: "So excited to see police officers participating in #LondonPride parade! This brings me hope that we can all learn to live together peacefully, despite our differences. #PoliceUnity #PrideAndUnity"
>
>     **ChatGPT**: "I can't believe it! Just found out some on-duty cops took part in #LondonPride parade! As a member of the LGBTQ+ community, I feel so betrayed and disrespected. It's not just about them wearing rainbows, it's about the constant abuse and discrimination we face from the police force. #Unacceptable #HomophobiaInUniform"
>
>     *The original claim does not mention the police and homophobia, but the ChatGPT claim expands on the original by including this information.*
>
> - **Uses the emotions from the prompt in more interesting ways** (example below)
>
>     **Original claim [emotion: sadness]**: "A study on vitamin D and Covid-19 shows an 80% reduction in need for ICU and a 60% reduction in deaths. The study demonstrates that the clear relationship between vitamin D and Covid mortality is causal."
>
>     **GPT3**: "Feeling so down about Covid-19. New study found an 80% reduction in the need for ICU & 60% reduction in deaths when Vitamin D is supplemented. Evidently, the link between Vitamin D & Covid-19 mortality is real. Time to up my Vitamin D intake! #StayHealthy #VitaminD #CovidMortality"
>
>     **ChatGPT**: "Just read this study on Vitamin D and Covid-19 and it's making me so sad! The study shows that if people have enough Vitamin D, their chances of dying from Covid goes down by 60%. And if they do end up in the ICU, they have an 80% chance of getting better. This means that so many lives could have been saved if we all just took our vitamins! It's heartbreaking to think about all the people who could have been saved if we just knew this sooner. #vitamind #covid19 #heartbroken"
>
>     *The GPT3 claim just says that they feel sad about Covid-19, but then the emotion is dropped entirely. The ChatGPT claim expands upon the emotion throughout the generated claim – ‘if we knew this sooner, more people could have been saved!’*
>
> - **Has more interesting phrasing and personal connections in general** (example below)
>
>     **Original claim [emotion: fear]**: "The mRNA Covid-19 vaccines are experimental."
>
>     **GPT3**: "I'm really scared by the #mRNA #Covid19 #vaccines bc they're still experimental. I don't know if it's safe for me to take one & the uncertainty is really overwhelming. #experimentalvaccine"
>
>     **ChatGPT**: "Just heard about the mRNA #Covid19 vaccines being experimental and I'm freaking out. I don't want to be a guinea pig for some new technology! This is my health we're talking about. How am I supposed to trust it? #scared #vaccinehesitancy"
>
>     *GPT3’s generated claim is very generic compared to ChatGPT – ‘I don’t know if it’s safe’ versus ‘I don’t want to be a guinea pig!’*
>
> On top of these results, the prices of ChatGPT’s API are much lower than those of GPT3.
>
> As regards the random samples of generated data, the evaluators were provided with different data. In particular, we created subsamples of data from the verdicts generated by our models (in the various configurations) while inferencing from the test set. The test set comprises data from FullFact (journalistic style) and from VerMouth (SMP-style and emotional-style data).

---

### Official Review · Reviewer_QBDx · 2023-08-12

**Soundness:** 3

**Excitement:**

4: Strong: This paper deepens the understanding of some phenomenon or lowers the barriers to an existing research direction.

**Paper Topic And Main Contributions:**

The paper develops a dataset of 12K factually incorrect claims. Each claim is also accompanied with a counter response that is designed to be polite and empathetic. The novelty of this dataset is that the claim-response pairs are generated for the informal style of writing on social media platforms and express a specific emotional register. These pairs are generated by an LLM but are modified by humans in a post-generation step. The paper also presents experiments that show better performance in some tasks for models that are trained on this new dataset.

**Questions For The Authors:**

* Q1. Why emotional styles yielded consitently longer claims and verdicts compared to SMP style?
* Q2. HTER is a measure for an individual pair. So in Table 4, is it averaged across the samples? According to the appendix, only ~19% samples required post-editing for SMP style, so is the average taken over the samples that required post-editing or over all the samples?
* Q3. Why is the RR > 1? Acc. to the Bertoldi 2013 and Cettolo 2014 papers, RR is normalized to be between 0 and 1.
* Q4. How many humans were involved in the reviewer stage? Were the same humans involved in session 1 and 2? I don't think there was much, if any discussion, about the human annotators in the paper about who the humans were, how they were chosen, what was their background, how they were trained, etc. Such questions are important to be addressed because past research (e.g., Sap et. al. 2019) have found that labels can be affected by annotator bias.

**Reasons To Accept:**

* The resource will be released publicly.
* The resource could be valuable for researchers in the community to build models that differentiate between claims and counterclaims.
* The generation of counterclaims with different styles and emotional registers can be used to study causal effects of presenting a counterclaim in a certain way.

**Reasons To Reject:**

* Details about the human annotators background is largely missing (see Q4)
* There are some inconsistencies in the calculation of certain metrics that should be clarified (see Q2 and Q3)

**Reproducibility:**

4: Could mostly reproduce the results, but there may be some variation because of sample variance or minor variations in their interpretation of the protocol or method.

**Reviewer Confidence:**

4: Quite sure. I tried to check the important points carefully. It's unlikely, though conceivable, that I missed something that should affect my ratings.

---

> ### Author Rebuttal · Authors · 2023-08-28
>
> Dear reviewer,
>
> We thank you for your valuable feedback. We will make sure to integrate the comments and suggestions you provided into the additional page provided for the final version of the paper upon acceptance. Hereafter, we will try to answer the questions and doubts you raised in your review.
>
> >*Q1. Why emotional styles yielded consistently longer claims and verdicts compared to SMP style?*
>
> Provided that all ChatGPT parameters were kept constant across generations and claim configurations, one reasonable explanation, derived from checking the outputs, is that the prompt used for emotional data was much more detailed - and contained more requirements - than the one used for SMP-style data (see Table 2). Thus, we can hypothesize that, for our task, ChatGPT tries to satisfy the multiple requirements (“SMP+emotional+personal style”) contained in the emotional prompt instruction by producing a longer output.
>
> >*Q2. HTER is a measure for an individual pair. So in Table 4, is it averaged across the samples? According to the appendix, only ~19% samples required post-editing for SMP style, so is the average taken over the samples that required post-editing or over all the samples?*
>
> HTER values were averaged over the entire samples under analysis (including the unmodified ones). We will specify this in the final version of the paper in case of acceptance and provide numbers also for modified ones alone.
>
> >*Q3. Why is the RR > 1? Acc. to the Bertoldi 2013 and Cettolo 2014 papers, RR is normalized to be between 0 and 1.*
>
> The reviewer is right but, in order to be in line with previous works, e.g.  Tekiroglu et al. 2020 (Table 1)  and Cettolo 2014 (Table 3), our RR values range between 0 and 100. We will make this clear in the final version of the paper.
>
> >*Q4. How many humans were involved in the reviewer stage? Were the same humans involved in sessions 1 and 2? I don't think there was much, if any discussion, about the human annotators in the paper about who the humans were, how they were chosen, what was their background, how they were trained, etc. Such questions are important to address because past research (e.g., Sap et al. 2019) have found that labels can be affected by annotator bias.*
>
> We apologise for the lack of details: annotators for sessions 1 and 2 were the same while the evaluation was carried out by different people to avoid any possible bias.
>
> We understand that the information regarding the annotators could have been enriched and we apologise for the lack of details. Specifically, the data generated with the LLMs were post-edited by two annotators: one last-year master's student and a Ph.D. student. Both of them are included as authors of the present work (in addition, the master's student got their university credit and the Ph.D. has a salary). Both annotators were extensively trained on the data and the topic of misinformation and automated fact-checking, as well as they had good experience with the communication style employed on the main social media platforms. In particular, before starting to work on the data they were properly trained for the task, by (i) providing relevant literature, (ii) walking them through the complete guidelines, (iii) providing and discussing several claim and verdict examples, and (iv) answering possible doubts. Moreover, weekly meetings were organized throughout the whole annotation campaign to discuss possible problems and doubts about specific post-editing instances. We will include these details in the additional page provided for the final version of the paper upon acceptance.

---

### Official Review · Reviewer_Emnp · 2023-08-12

**Soundness:** 4

**Excitement:**

4: Strong: This paper deepens the understanding of some phenomenon or lowers the barriers to an existing research direction.

**Paper Topic And Main Contributions:**

The paper proposes a new dataset consisting of claims-responses pairs to counter misinformation. The authors have leveraged the author-reviewer pipeline to create the dataset and performed both automatic and human evaluations.

**Reasons To Accept:**

-	Explanations for each step of dataset generation provided including assumptions, reasons, examples and references.
-	Clear definitions for terms
-	Well-formatted paper
-	Interesting novel approach to counter misinformation


**Reasons To Reject:**

-	Annotator selection criteria and salary not mentioned.
-	Though 3 human panel for evaluations is common in the computer science field, this use case doesn’t seem to be representative of how larger social groups would perceive the responses to misinformation.


**Reproducibility:**

2: Would be hard pressed to reproduce the results. The contribution depends on data that are simply not available outside the author's institution or consortium; not enough details are provided.

**Reviewer Confidence:**

3: Pretty sure, but there's a chance I missed something. Although I have a good feel for this area in general, I did not carefully check the paper's details, e.g., the math, experimental design, or novelty.

---

> ### Author Rebuttal · Authors · 2023-08-28
>
> Dear Reviewer,
>
> We deeply thank you for the feedback you provided.
>
> We understand that the information regarding the annotators could have been enriched and we apologise for the lack of details. Specifically, the data generated with the LLMs were post-edited by two annotators: one last-year master's student and a Ph.D. student. Both of them are included as authors of the present work (in addition, the master's student got their university credit and the PhD has a salary). Both annotators were extensively trained on the data and the topic of misinformation and automated fact-checking, as well as they had good experience with the communication style employed on the main social media platforms. In particular, before starting to work on the data they were properly trained for the task, by (i) providing them with relevant literature, (ii) walking them through the complete guidelines, (iii) providing and discussing with them several claims and verdicts examples, and (iv) answering possible doubts. Moreover, weekly meetings were organized throughout the whole annotation campaign to discuss problems and doubts about post-editing that might have arisen. We will include these details in the additional page provided for the final version of the paper upon acceptance.
>
> Regarding the human evaluators, we believe that a study representative of how larger social groups would perceive the responses to misinformation is a relevant topic but out of the scope of the present work. In our evaluation we wanted to check how the models' generations were perceived at first glance, trying to assess the quality of the generations both in in-domain and cross-domain settings. Further ad hoc studies employing a larger number of evaluators are indeed needed.
>
> Finally, as already mentioned in the paper, upon acceptance we will make publicly available the resource as well as the accompanying code for reproducibility purposes.

---

### Meta-Review · Area_Chair_K3mW · 2023-09-22

**Recommendation:** 5

**Metareview:**

This study proposes a dataset consisting of approximately 12K claim-response pairs, which are linked to debunked articles. Each claim in the dataset is accompanied by an emotional response, aimed at countering misinformation.

All reviewers appreciated the contribution of the dataset. However, there are some concerns. Reviewers Emnp, QBDx, and NTyP mentioned the lack of details regarding the annotators, which the authors addressed in the rebuttal. Please include this information in the paper while making revisions. Additionally, an important point raised by Reviewer LBT1 regarding LLM hallucinations should be addressed in the paper.

---

### Decision · Program_Chairs · 2023-10-07

**Decision:**

Accept-Main

**Comment:**

This study proposes a dataset consisting of approximately 12K claim-response pairs, which are linked to debunked articles. Each claim in the dataset is accompanied by an emotional response, aimed at countering misinformation.

All reviewers appreciated the contribution of the dataset. However, there are some concerns. Reviewers Emnp, QBDx, and NTyP mentioned the lack of details regarding the annotators, which the authors addressed in the rebuttal. Please include this information in the paper while making revisions. Additionally, an important point raised by Reviewer LBT1 regarding LLM hallucinations should be addressed in the paper.